# A live attenuated-vaccine model confers cross-protective immunity against different species of the Leptospira genus

Elsio A Wunder[1,2]*, Haritha Adhikarla[1], Camila Hamond[1], Katharine A Owers Bonner[1], Li Liang[3], Camila B Rodrigues[3,4], Vimla Bisht[1], Jarlath E Nally[5], David P Alt[5], Mitermayer G Reis[2], Peter J Diggle[6], Philip L Felgner[3], Albert Ko[1,2]

[1]Department of Epidemiology of Microbial Diseases; Yale School of Public Health, New Haven, United States; [2]Gonçalo Moniz Institute, Oswaldo Cruz Foundation; Brazilian Ministry of Health, Salvador, Brazil; [3]Department of Medicine, Division of Infectious Disease; University of California Irvine, Irvine, United States; [4]Institute of Technology in Immunobiology, Oswaldo Cruz Foundation; Brazilian Ministry of Health, Rio de Janeiro, Brazil; [5]Infectious Bacterial Diseases Research Unit, National Animal Disease Center, Agricultural Research Service; United States Department of Agriculture, Ames, United States; [6]CHICAS, Lancaster Medical School; Lancaster University, Lancaster, United Kingdom

**Abstract** Leptospirosis is the leading zoonotic disease in terms of morbidity and mortality worldwide. Effective prevention is urgently needed as the drivers of disease transmission continue to intensify. The key challenge has been developing a widely applicable vaccine that protects against the >300 serovars that can cause leptospirosis. Live attenuated mutants are enticing vaccine candidates and poorly explored in the field. We evaluated a recently characterized motility-deficient mutant lacking the expression of a flagellar protein, FcpA. Although the *fcpA*⁻ mutant has lost its ability to cause disease, transient bacteremia was observed. In two animal models, immunization with a single dose of the *fcpA*⁻ mutant was sufficient to induce a robust anti-protein antibodies response that promoted protection against infection with different pathogenic Leptospira species. Furthermore, characterization of the immune response identified a small repertoire of biologically relevant proteins that are highly conserved among pathogenic Leptospira species and potential correlates of cross-protective immunity.

*For correspondence:
elsio.wunder@yale.edu

## Introduction

Leptospirosis is caused by a genetically and antigenically diverse group of spirochetes of the Leptospira genus (*Picardeau, 2017*; *Ko et al., 2009*; *Adler, 2015*). Currently the Leptospira genus comprises 64 species with more than 300 serovars, with 17 of those species containing strains that can potentially cause severe disease in humans and animals (*Casanovas-Massana et al., 2020*; *Vincent et al., 2019*). A broad range of mammalian reservoirs harbor the spirochete in their renal tubules shedding the bacteria in their urine for long periods of time (*Thibeaux et al., 2017*; *Xu et al., 2016*). Leptospirosis is an environmentally transmitted disease with great health and economic impact in both humans and animals, for which the primary mode of transmission to humans is through contact with contaminated water or soil (*Casanovas-Massana et al., 2018a*).

Although Leptospira has a worldwide distribution, the large majority of the burden occurs in the world's most impoverished populations (*Costa et al., 2015*), where the rapid growth of urban slums worldwide has created conditions for rat-borne transmission. The disease causes life-threatening

**eLife digest** Leptospirosis is a life-threatening disease with flu-like symptoms that is caused by bacteria known as *Leptospira*. It is more common in warmer regions with high rainfall, especially in impoverished areas. The disease is spread in the urine of animals such as rodents, farm animals or dogs. Humans and other animals can get leptospirosis when they come in contact with urine-contaminated water and soil.

Current measures to control leptospirosis are largely ineffective. Although a vaccine is available for animals, it only protects against a few types of the 300 disease-causing *Leptospira* bacteria. It also fails to stop the bacteria from colonizing the kidneys of the infected animals, which means that vaccinated animals can still spread disease.

Previous research has shown that inactivating a protein called FcpA, which is necessary for *Leptospira* bacteria to move, can stop them from infecting hamsters. Moreover, when these animals were exposed to the mutant bacteria, they did not get sick nor developed the disease. Here, Wunder et al. tested whether bacteria lacking the FcpA protein could be used as an attenuated vaccine. This form of vaccine contains live bacteria that have been modified to become harmless but are able to train the immune system to produce a long-lasting immune response against the invaders.

The results showed that a single dose of the vaccine was enough to prevent hamsters and mice from dying of leptospirosis. It also worked against several types of *Leptospira* and could stop them from colonizing mice kidneys. Moreover, Wunder et al. found that the immune system targeted specific proteins that were common to various types of *Leptospira*, which may explain the broad spectrum of protection the vaccine offered.

Rapid urbanization and climate change are among the main drivers of leptospirosis. An effective vaccine for this disease would reduce the public health burden by providing protection against leptospirosis and by reducing the spread of the disease. A next step will be to ensure the mutant *Leptospira* are safe to use in animals and potentially humans.

---

manifestations such as Weil's disease (*Ko et al., 1999*; *Adler, 2015*) and leptospiral pulmonary hemorrhage syndrome (LPHS) (*Gouveia et al., 2008*). A recent study estimated that leptospirosis causes 1.03 million cases and 58,900 deaths each year (*Costa et al., 2015*). Case fatality for Weil's disease and LPHS is >10 and >50%, respectively, despite aggressive supportive care (*Gouveia et al., 2008*). These estimates place leptospirosis as a leading zoonotic cause of morbidity and mortality worldwide. The burden of leptospirosis will increase as climate and land use change continues to evolve and the world's slum population doubles to 2 billion by 2025 (*UN-HABITAT, 2003*).

The public health priority is therefore prevention of leptospirosis before severe complications develop. However, there is no effective control for leptospirosis and safe and efficacious vaccines are not available for human use (*Ko et al., 2009*; *Picardeau, 2017*). China and Cuba use whole-cell vaccines in humans (*Yan et al., 2003*; *Martínez et al., 2004*), but they are not licensed to be used elsewhere. Whole-cell vaccines are widely used for veterinary purposes but have significant limitations, since immunity is of short duration and predominantly humoral against LPS, which are serovar-specific moieties. Multivalent vaccines are unable to achieve sufficient coverage against the spectrum of serovars that are important for animal and human health (*Adler, 2015*). Research has thus focused on characterizing surface-associated and host-expressed proteins as sub-unit vaccine candidates (*Adler, 2015*; *Ko et al., 2009*). To date, these conventional approaches have not yielded candidates and attempts have failed to identify a universal, widely applicable vaccine.

Current control measures have been uniformly ineffective in addressing the large human and animal global health burden due to leptospirosis, especially in developing countries. Given the limitations of the whole-cell vaccines available and the ineffective attempts to identify protein vaccine candidates (*Adler, 2015*), an attenuated-vaccine approach remains a feasible strategy. Attenuation of Leptospira virulence has been long-recognized yet poorly understood phenomenon (*Adler, 2015*; *Srikram et al., 2011*). Until recently, the inability to produce well-defined mutants has preempted efforts to identify a safe and efficacious attenuated-vaccine. However, current advances in genomic tools and whole-genome sequencing data for Leptospira (*Thibeaux et al., 2017*; *Picardeau, 2017*)

have circumvented this limitation and some promising results have been shown (*Srikram et al., 2011*; *Murray et al., 2018*).

Recently, our group identified and characterized a novel flagellar protein in the Leptospira genus involved in the composition of the sheath of the leptospiral flagella, Flagella-coil protein A (FcpA). A mutant deficient in the *fcpA* gene lost its ability to produce translational motility and to penetrate mucous membranes, resulting in loss of kidney colonization and lethality in the hamster model of leptospirosis. Although highly attenuated in the hamster model, a needle inoculation of the mutant produced a transient bacteremia prior to clearance by the host immune response (*Wunder et al., 2016a*). In the present study, we evaluated the *fcpA*- motility-deficient mutant as a potential candidate for a live attenuated-vaccine that could provide a major public health benefit and opportunity to leverage One Health approaches.

## Results

### A motility-deficient strain as an attenuated-vaccine candidate

We characterized a previously unidentified flagellar sheath protein (FcpA) that was essential for translational motility and thus for virulence (*Wunder et al., 2016a*). Despite the phenotype of complete attenuation, we observed that the L1-130 *fcpA*- mutant caused a transient systemic infection, which was cleared 7 days after intraperitoneal inoculation of $10^8$ leptospires in hamsters (*Wunder et al., 2016a*). In this study, after inoculation of $10^7$ leptospires using the subcutaneous route of infection in hamsters, we detected the presence of DNA of the mutant by qPCR in all the tissues tested, with the exception of the brain (*Figure 1A*). These results were similar to those observed previously, with the wild-type reaching higher number of leptospires in all tissues analyzed, leading to the death or euthanasia of the animals due to clinical signs of disease 5–7 days after infection. In comparison, the signal for the *fcpA*- mutant strain was undetectable after 7 days with all inoculated animals surviving with no detectable leptospires in either kidney or blood, measured by qPCR and culture. Similarly, no detectable signal was observed for the animals immunized with the L1-130 heat-killed strain (*Figure 1A*). We also tested the *fcpA*- mutant in the mouse model using different doses of infection (*Figure 1B*). Although the dose of the wild-type strain was not enough to produce disease and lethality on infected mice, all animals were colonized and the presence of the leptospiral DNA in blood was detectable until the fifteenth day after infection (*Figure 1B*). Furthermore, no dose of the *fcpA*- mutant caused colonization (data not shown) and there was a significant difference in the magnitude of dissemination of the mutant in the blood compared to the wild type (*Figure 1B*). DNA signal of the *fcpA*- mutant was only observed in the blood of animals infected with doses of $10^7$ and $10^5$ until the 13th and 8th day after infection, respectively. Taken together, these results indicate that although the *fcpA*- mutant is attenuated in both the hamster and mouse model, there is a hematogenous dissemination of this mutant, identified by detection of its DNA. The mutant appears to be cleared by the immune system before it results in observable disease or death of the animals. Furthermore, we observed in the mouse model that the dissemination of the mutant is dose dependent. However, it is important to notice that although we do not see any signal of the mutant in doses equal or lower to $10^3$ leptospires, the theoretic limit of detection of the qPCR assay used here (*Wunder et al., 2016a*; *Stoddard et al., 2009*) is 100 leptospires/mL of blood which can result in false negative results.

### Model for cross-protective immunity to leptospirosis

We hypothesized that the transient infection produced by the *fcpA*- mutant induces cross-protective responses, given previous findings (*Wunder et al., 2016a*; *Srikram et al., 2011*). Immunization with a single dose of the *fcpA*- mutant (*Figure 2A*) conferred complete protection against mortality in hamsters from infection with homologous and heterologous serovars (*Figure 2B* and *Supplementary file 2*). In contrast, immunization with heat-killed leptospires conferred partial protection to the homologous but not against the heterologous serovar (*Figure 2B* and *Supplementary file 2*). Heat-killed bacterins can give a high protection level against an homologous challenge (*Adler, 2015*), but usually the protocol for vaccination includes at least a second dose of the vaccine. Our poor results here with the heat-killed vaccine, especially for the homologous challenge (*Figure 2B* and *Supplementary file 2*), might be due to the lack of a vaccine boost. For the

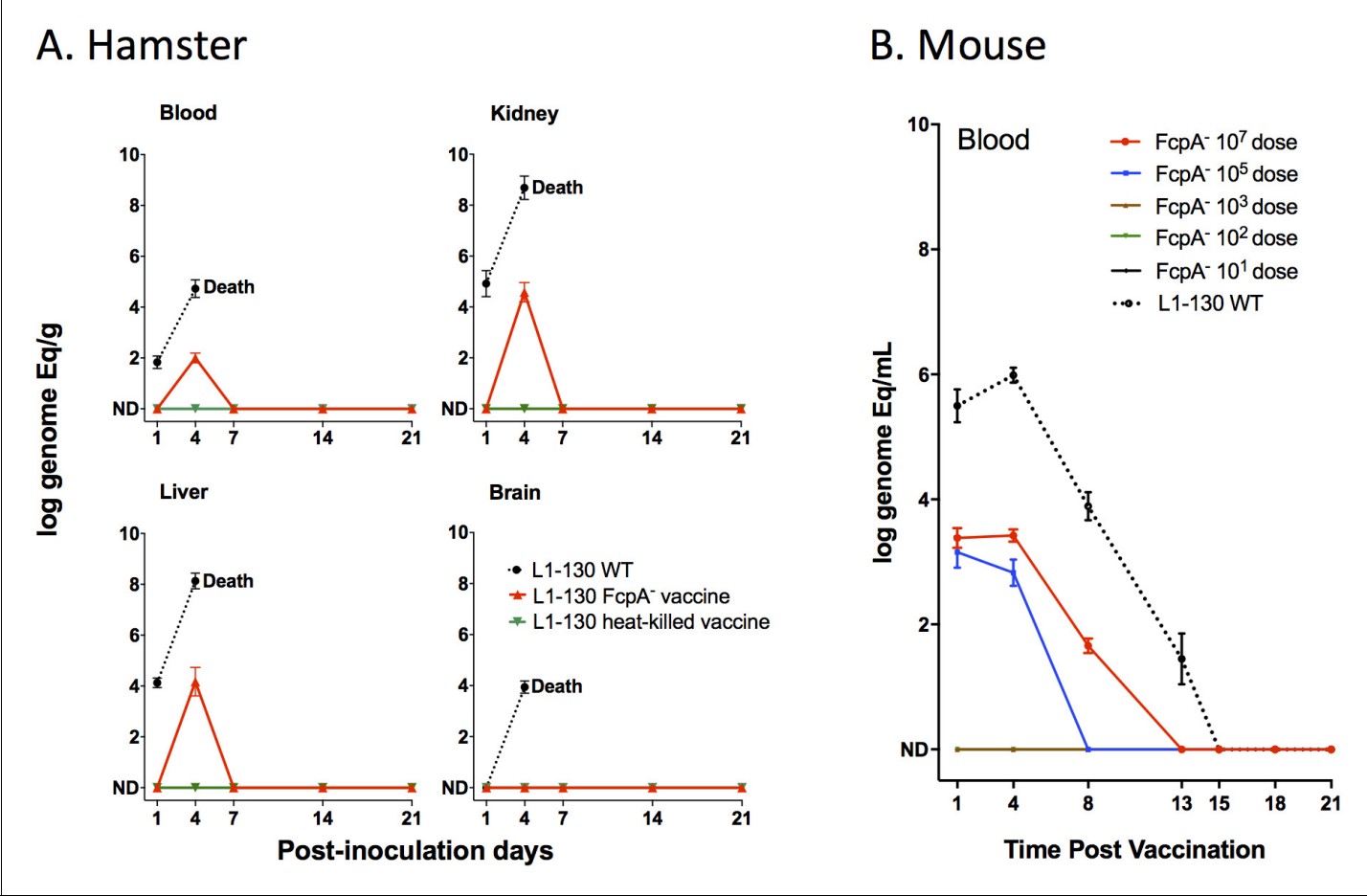

**Figure 1.** Dissemination of L1-130 *fcpA⁻* mutant in animal tissues. (**A**) Kinetics of infection of L1-130 WT, L1-130 *fcpA⁻* vaccine, and L1-130 heat-killed vaccine in blood, kidney, liver, and brain of hamsters after inoculation with $10^7$ bacteria. All animals infected with WT strain died between 5 and 6 days post-infection; (**B**) Kinetics of infection of L1-130 WT ($10^7$ leptospires) and L1-130 *fcpA⁻* attenuated-vaccine (dose range from $10^7$ to $10^1$ leptospires) in blood of mouse. Results are expressed by logarithmic genome equivalent per gram or milliliter of tissues with mean and standard deviation. All doses were inoculated by subcutaneous route in both models.

The online version of this article includes the following source data for figure 1:

**Source data 1.** Raw data for dissemination experiments in hamster and mouse.

purpose of evaluating the efficacy of the attenuated-vaccine after a single dose, we decided to keep a standard protocol for vaccination and thus using only one dose of the heat-killed vaccine as well. It is important to mention that the strain Hardjo 203 was described to cause only colonization in the hamster model infected by intraperitoneal route (*Zuerner et al., 2012*). However, in our LD₅₀ experiments using the conjunctival route we reproducibly observed 25% death rate when using the conjunctival route (*Supplementary file 1*). Furthermore, in the non-vaccinated group we observed an overall death rate of 21.4% after challenge with the strain Hardjo 203, but no deaths in the vaccinated group, which explains the wide 95% CI range (*Figure 2B* and Table S2).

Protection against renal colonization was only observed in 80% of the animals immunized with *fcpA⁻* mutant after homologous infection. Heterologous infection gave varying levels of protection, from 0% to 35.7% (*Figure 2D* and *Supplementary file 2*). Hamsters are highly susceptible to leptospirosis (*Haake, 2006*), so the finding that the attenuated strain conferred partial protection against colonization was not unexpected. To understand the efficacy of the *fcpA⁻* mutant vaccine to protect against colonization, we tested different doses of immunization using the mouse model against heterologous infection. Our results indicate that the protection conferred by the *fcpA⁻* mutant is dose dependent. Against death, the vaccine conferred 100% protection up to a dose of $10^3$ leptospires of the *fcpA⁻* mutant (*Figure 2C* and *Supplementary file 3*), but a dose as high as $10^7$ leptospires was

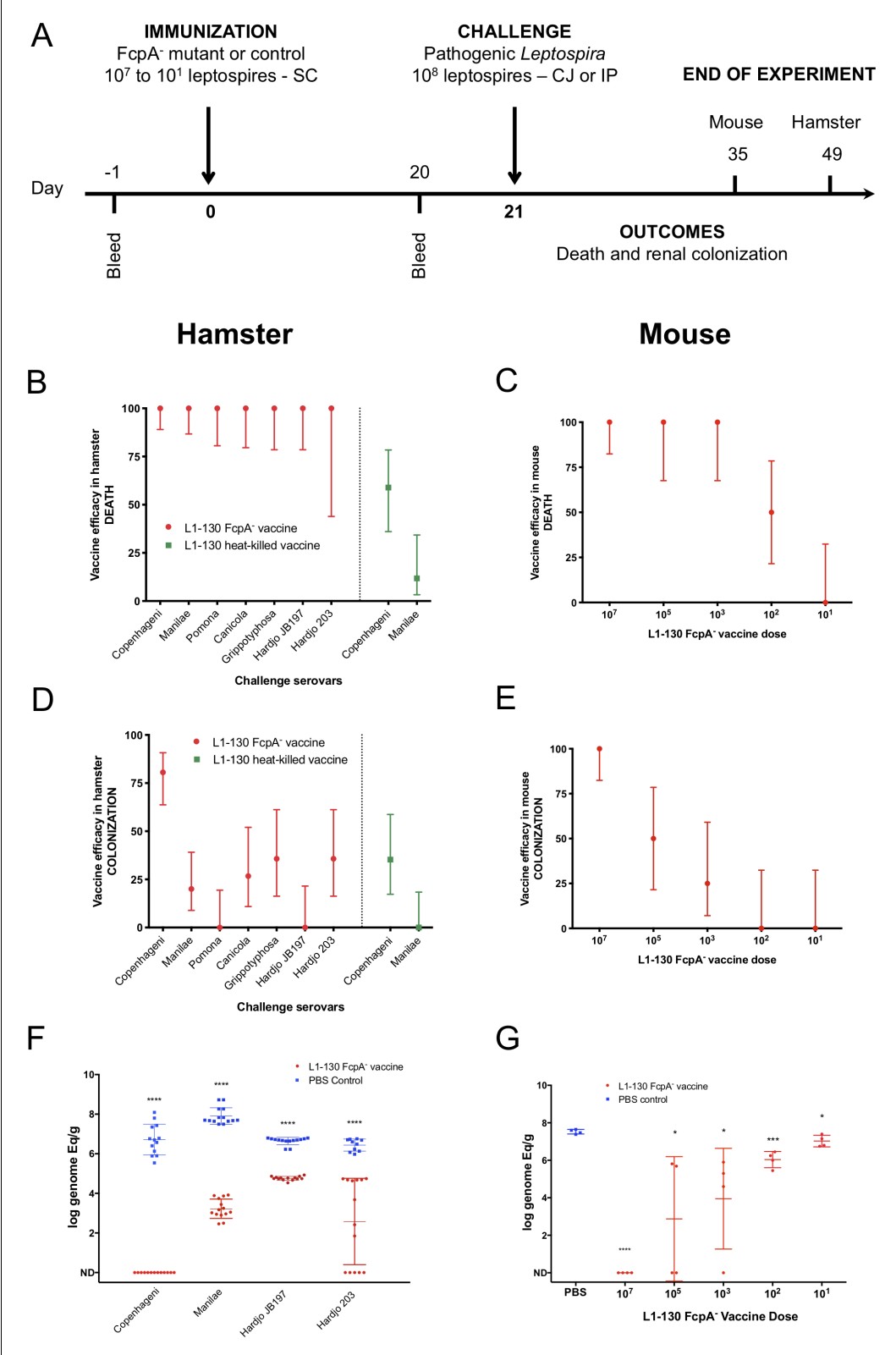

**Figure 2.** Efficacy of L1-130 *fcpA⁻* attenuated-vaccine model. Animals were vaccinated with a dose of $10^7$ leptospires (hamsters) or a range of doses from $10^7$ to $10^1$ leptospires (mice) by subcutaneous (SC) route. Animals were bled the day before immunization (day −1) and day 20 post-immunization (**A**). Hamsters were challenged by conjunctival route with either the homologous strain or different heterologous strains. Mice were challenged by intraperitoneal route with the heterologous serovar Manilae of *L. interrogans*. By combining all vaccine experiments performed, efficacy of the vaccine

*Figure 2 continued on next page*

*Figure 2 continued*

against death and colonization was evaluated for hamsters (**B and D**) and mice (**C and E**) and represented by percentage and 95% CI based on frequency of outcomes compared to PBS-immunized animals. Hamster experiment are showing the results after vaccination with the *fcpA*⁻ attenuated-vaccine (red) and heat-killed vaccine (blue). Bacterial load in the kidney was measured by qPCR in hamsters (**F**) and mice (**G**) and compared between PBS-immunized animals (blue) and animals immunized with *fcpA*⁻ attenuated-mutant (red). Results are expressed in logarithmic genome equivalents per gram of renal tissue with mean and standard deviation. Asterisk symbols represent statistical significance calculated by t-test: *p<0.01, ***p<0.0001. See also *Supplementary files 2* and *3*.

The online version of this article includes the following source data for figure 2:

**Source data 1.** Raw data for qPCR experiments in hamster and mouse.

necessary to obtain 100% protection against colonization (*Figure 2E* and *Supplementary file 3*). Furthermore, our quantitative analyzes of renal colonization showed that although the *fcpA*⁻ mutant cannot promote complete protection, there is a significant reduction of the burden of the disease both in hamster after heterologous infection (*Figure 2F* and *Supplementary file 2*) and in lower doses of the vaccine in the mouse model, which also revealed a dose-dependent phenotype (*Figure 2G* and *Supplementary file 3*).

These findings indicate that a single dose of a live attenuated-vaccine elicits cross-protective immunity against serovars belonging to *L. interrogans*, *L. kirschneri*, and *L. borgpetersenii*, the species which encompasses the majority of serovars of human and animal health importance.

## Antibodies against Leptospira proteins as a correlate for the cross-protective immunity

The *fcpA*⁻ attenuated-vaccine induced a weak agglutinating antibodies response to the homologous serovar, Copenhageni, and undetectable microscopic agglutination test (MAT) titers against heterologous serovars, both in hamsters (*Figure 3A*) and mice (*Figure 3C*). Furthermore, in the mouse

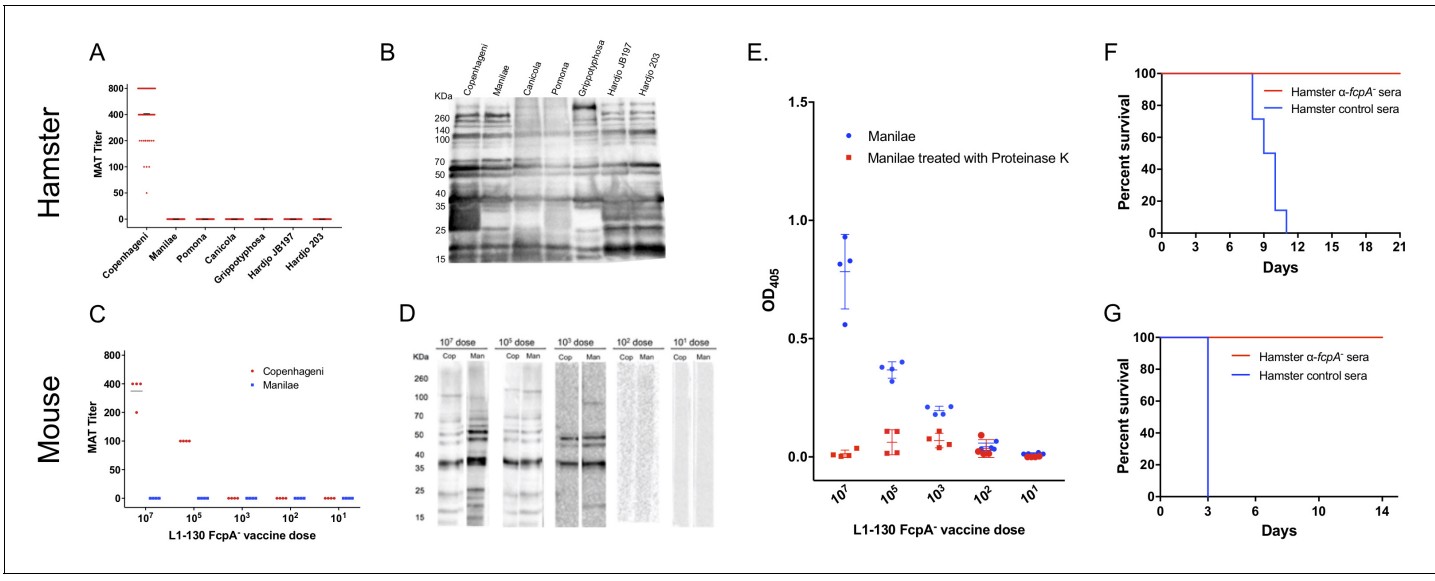

**Figure 3.** Immunogenicity and correlates of immunity for L1-130 *fcpA*⁻ attenuated-vaccine model. Individual sera of hamsters and mice were obtained after 20 days post-vaccination by a subcutaneous (SC) dose of 10⁷ leptospires (hamsters) or a range of doses from 10⁷ to 10¹ leptospires (mice) of the attenuated-vaccine. Microscopic agglutination test (MAT) (**A and C**) and western blot (**B and D**) were performed adopting as antigen all the strains used for challenged in both hamster and mice, respectively. Mice sera was additionally tested using an ELISA assay (**E**) adopting whole-cell extract of serovar Manilae with (red) and without (blue) Proteinase K treatment as antigen. Furthermore, a pool of hamster immune-sera vaccinated with a dose of 10⁷ leptospires of *fcpA*⁻ attenuated-vaccine was used for passive transfer experiments. 2 mL or 0.5 mL of sera was passively transfer to naïve hamsters (**F**) or mice (**G**), respectively, followed by challenge with a dose of 10⁸ leptospires of heterologous serovar Manilae by conjunctival (CJ) or intraperitoneal (IP) route, respectively. Results are expressed in a survival curve of animals passively transferred with *fcpA*⁻ anti-sera (red) and control hamster sera (blue).

The online version of this article includes the following source data for figure 3:

**Source data 1.** Raw data for microscopic agglutination test (MAT) and passive transfer experiments in hamster and mouse, and ELISA data in mouse.

model, agglutinating antibodies were only measurable with a dose of at least $10^5$ leptospires (*Figure 3C*). In contrast, a single dose of the *fcpA⁻* mutant was able to induce a robust immune response against leptospiral proteins, recognizing proteins across the different species of Leptospira used in the hamster model (*Figure 3B*) and the heterologous strain used in the mouse model with a dose of at least $10^3$ leptospires (*Figure 3D*). In addition, the presence of detectable antibodies measured by ELISA correlates with the highest dose that induced 100% protection against death in the mouse model ($10^3$ leptospires), and there is a decrease on the OD for all doses when the Manilae antigen was treated with proteinase K (*Figure 3E*). Furthermore, our passive transfer experiments using hamster-immune sera against *fcpA⁻* attenuated-vaccine conferred 100% protection against heterologous lethal infection in hamsters (*Figure 3F*) and mice (*Figure 3G*). Taken together, these results indicate that anti-Leptospira protein antibodies, and not agglutinating antibodies, are the correlate of vaccine-mediated cross-protective immunity.

## Highly conserved seroreactive proteins as potential targets for eliciting cross-protective responses

We characterized the antibody response to the attenuated-vaccine using a downsized proteome array of 660 and 330 ORFs for hamster and mouse sera, respectively. We identified a total of 133 (*Figure 4A*) and 56 (*Figure 4B*) protein targets on our analysis of hamsters (Hamster $10^7$) and mice (Mouse $10^7$) respectively, immunized with a dose of $10^7$ leptospires and a total of 13 protein targets (*Figure 4C*) on our analysis of mouse immunized with different doses of the attenuated-vaccine (Mouse all). The reason to analyze the mouse results separately was based on the fact that a dose of $10^7$ leptospires of the attenuated-vaccine was able to give 100% cross-protection against lethality and colonization (*Figure 2C and E*). When combined, these three different analyses resulted in a total of 154 unique protein targets (*Figure 4D* and *Supplementary file 4*). Of those, 55% (85) have no prediction of localization and 23% (36), 14% (21), and 8% (12) have a prediction to be cytoplasmic membrane-associated, outer membrane proteins (OMP), and cytoplasmic, respectively (*Figure 4—figure supplement 1A*). Enrichment analysis showed a 5.0-fold (p=4.51E-10) and 1.8-fold (p=2.92E-04) enrichment for OMP and cytoplasmic membrane-associated, respectively (*Figure 4—figure supplement 1B*). In contrast, cytoplasmic proteins were 0.3-fold (p=2.91E-10) underrepresented in reactive antigens groups (*Figure 4—figure supplement 1B*).

Clusters of orthologous groups (COGs) of proteins were widely represented in those targets (*Supplementary file 4*), with at least one protein in each of the 18 functional categories. The COGs with higher representation were general function prediction only (R), cell wall/membrane/envelope biogenesis (M), intracellular trafficking, secretion, and vesicular transport (U), and cell motility (N) with 19, 17, 16, and 14 proteins, respectively. However, in addition to the 11 protein targets assigned as function unknown (S), the vast majority of the proteins had no COG assigned (59) (*Figure 4—figure supplement 1C*). Enrichment analysis identified proteins with predicted COG-U, COG-N, and COG-M function as highly enriched among the reactive antigens, by 4.9-fold (p=2.27E-07), 3.1-fold (p=8.35E-05), and 1.6-fold (p<0.05), respectively (*Figure 4—figure supplement 1D*). Furthermore, proteins predicted to be involved in signal transduction mechanisms (COG-T) and in amino acid transport and metabolism (COG-E) were significantly underrepresented in reactive antigens, by 0.4-fold (p=0.016) and 0.3-fold (0.02), respectively (*Figure 4—figure supplement 1D*). Taken together, the enrichment analysis validates our approach to identify biologically relevant protein candidates for a cross-protective vaccine.

We were able to narrow down the identified 154 proteins to 41 protein targets based on their relationship among the three different groups of the proteome array's analysis (*Figure 5* and *Figure 5—figure supplement 1*). Seven proteins were identified in all groups (*Figures 4D* and *5* and *Supplementary file 4*, red) and 31 proteins were identified in both hamster and mouse vaccinated with a dose of $10^7$ leptospires of the attenuated-vaccine (*Figures 4D* and *5* and *Supplementary file 4*, yellow). Furthermore, we identified three extra proteins identified in the group of mice immunized with different doses, two between the group of mice immunized with a dose of $10^7$ leptospires (*Figures 4D* and *5* and *Supplementary file 4*, green) and one extra protein between the group of hamsters immunized with a dose of $10^7$ leptospires (*Figures 4D* and *5* and *Supplementary file 4*, blue). Hamster and mice immune sera were highly reactive to the majority of the 41 proteins (*Figure 5*), in contrast to the low reactivity for the control sera and animals vaccinated with the heat-killed vaccine (*Figure 5—figure supplement 1*), indicating the ability of the attenuated-vaccine to

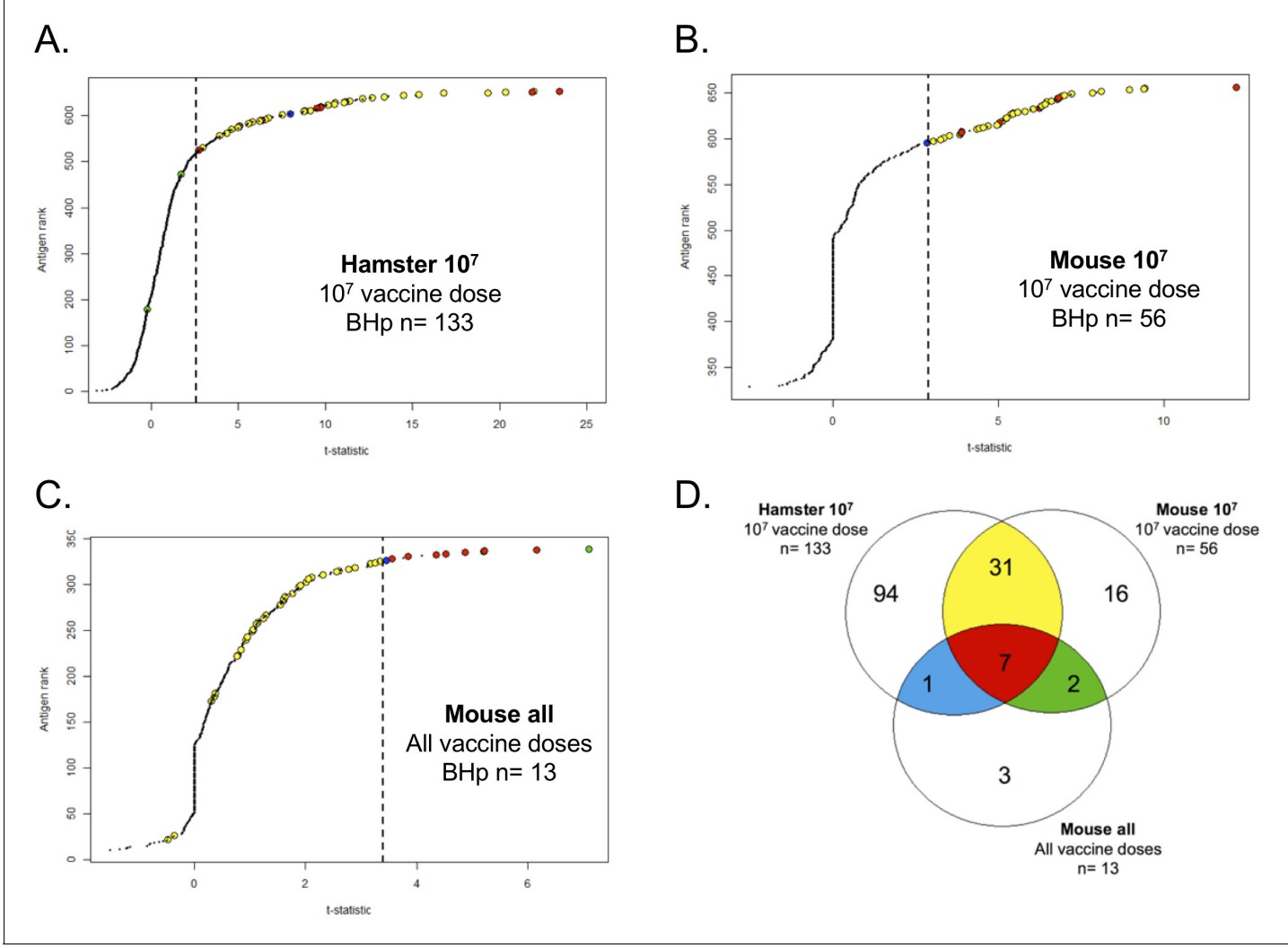

**Figure 4.** Proteome array analysis of immune-sera against L1-130 *fcpA⁻* attenuated-vaccine. Using statistical modeling we calculated the t-statistics value for each individual antigen used in the proteome array (660 for hamster and 330 for mice) based on three groups: the contrast between vaccinated and unvaccinated hamsters (A) or mice (B) using a vaccine dose of $10^7$ leptospires; the dose–response relationship for each antigen on mice (C) vaccinated with a range of doses from $10^7$ to $10^1$ leptospires of the attenuated-vaccine. Results are ranked based on individual t-statistics values for each antigen, and the dashed line represents the selection point for the antigens based on Bhp-test. The Venn-diagram (D) shows the relationship of all the 154 antigens identified in the three groups. The subgroups of antigens selected for further characterization are highlighted in color. See *Figure 5*. The online version of this article includes the following source data and figure supplement(s) for figure 4:

**Source data 1.** Raw data of proteome array experiments in hamster and mouse.
**Figure supplement 1.** In-silico analysis of protein targets.
**Figure supplement 2.** Mouse dose–response relationship.

induce immunity against leptospiral proteins. We identified plausible vaccine candidates among these 41 seroreactive proteins (*Figure 5*), which included six OMPs and known putative virulence factors such as LipL32, LipL41, and Lig proteins (*Ko et al., 2009*; *Picardeau, 2017*), providing supportive evidence for using proteome arrays to identify such proteins. Not surprisingly, 40% of those targets are identified as hypothetical proteins with no described function. However, the majority (70%) have high amino acid sequence identity (>80%) among their respective orthologs in all the 13 pathogenic Leptospira species analyzed (*Figure 5*), and therefore may be targeted for eliciting cross-protective responses. Moreover, sera from confirmed patients with acute leptospirosis reacted with 17 of the 41 Leptospira proteins recognized by sera from animals immunized with the attenuated-vaccine (*Figure 5—figure supplement 1*).

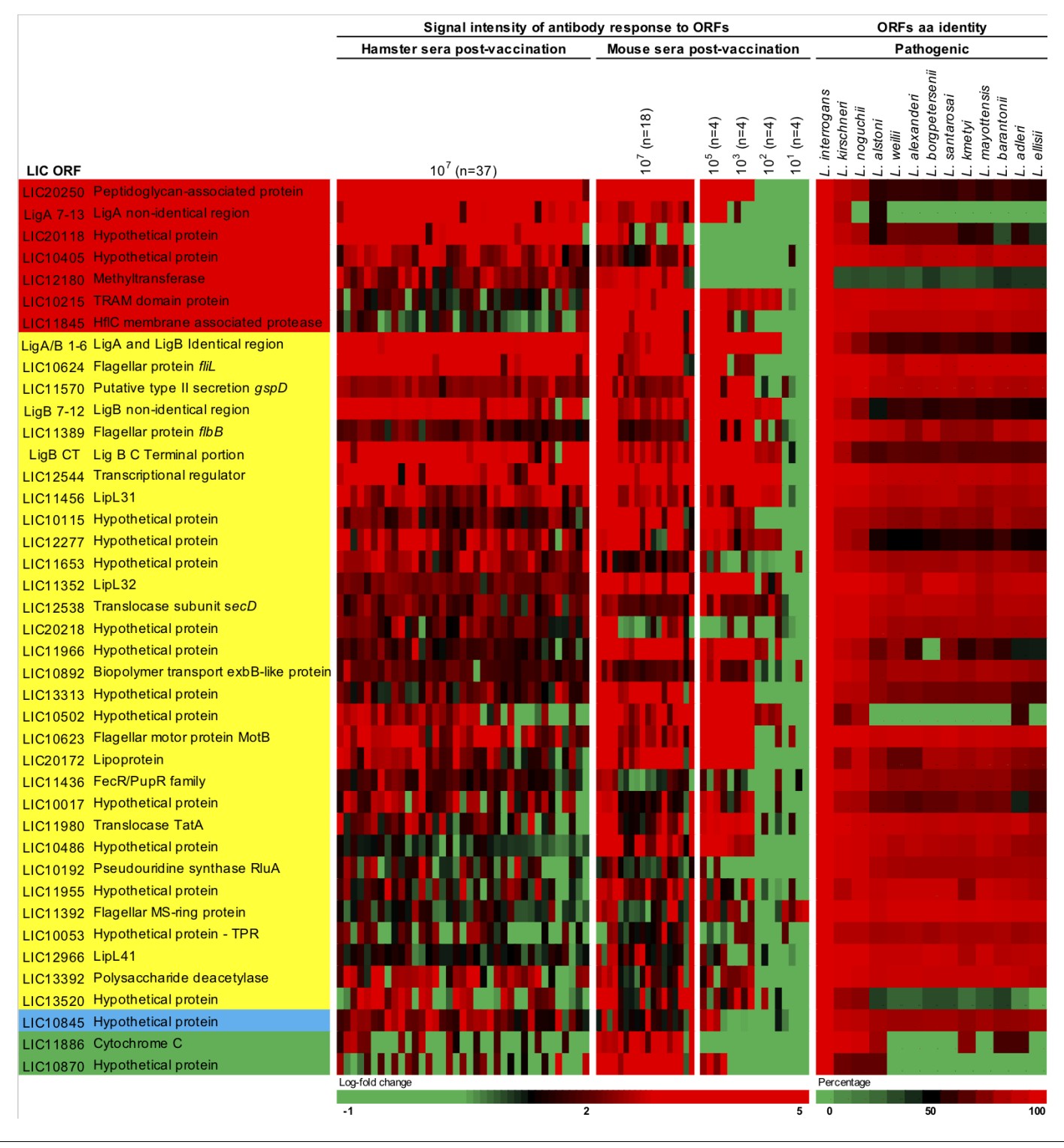

**Figure 5.** Heat-map of 41 seroreactive proteins recognized by hamsters and mice immunized with attenuated L1-130 *fcpA⁻* attenuated-vaccine. Proteins were selected based on the groups depicted on *Figure 4* and *Supplementary file 4*: present in all three groups of analysis (red), present in both hamster and mice immunized with $10^7$ leptospires (yellow), present in both hamsters immunized with $10^7$ leptospires and mice immunized with a dose range (blue), and present in both mice immunized with $10^7$ leptospires and mice immunized with a dose range (green). The proteins are identified by their *L. interrogans* serovar Copenhageni ORF number and the heat-map shows the signal intensity of antibody response (based on log-fold change) in all animals vaccinated with the *fcpA⁻* mutant used for this analysis (37 hamsters and 34 mice). Right panel shows amino acid sequence identity of respective ORFs among a representative of all pathogenic Leptospira species.

*Figure 5 continued on next page*

*Figure 5 continued*

The online version of this article includes the following source data and figure supplement(s) for figure 5:

**Source data 1.** Raw data of amino acid identity of 41 selected protein targets among different Leptospira species.
**Figure supplement 1.** Complementary heat-map of 41 seroreactive proteins recognized by hamsters and mice immunized with attenuated L1-130 *fcpA⁻* attenuated-vaccine.

## Discussion

In this proof-of concept study, we showed the efficacy of the *fcpA⁻* attenuate-vaccine in preventing both death and renal colonization in animal models. Live attenuated-vaccines are one of the most universal vaccination technologies used for the prevention of important bacterial diseases like tuberculosis, cholera, and salmonellosis and viruses like yellow fever, influenza, and zoster (*Minor, 2015*; *Detmer and Glenting, 2006*). Despite the risks and the hurdles to identify the best candidates, including the best balance between attenuation and immunogenicity, research in this field continues to increase (*Detmer and Glenting, 2006*; *Loessner et al., 2008*). An effective vaccine against leptospirosis would provide interdependent health and societal benefits by preventing transmission and disease in livestock and domestic animals and reducing the risk of spill-over infections in humans.

The transient bacteremia produced by needle inoculation of a single dose of the *fcpA⁻* mutant was sufficient to induce robust and cross-protective immunity in two different animal models of leptospirosis. The first live attenuated-vaccine for leptospirosis with a defined mutation has been recently described (*Srikram et al., 2011*; *Murray et al., 2018*) showing cross-protective immunity in hamsters and the potential for this approach. These authors made the important finding that immunization with an attenuated, LPS-deficient strain of serovar Manilae conferred varied levels of cross-protection against different species after one or two doses of the attenuated-vaccine. In our experiments, immunization with a single dose of the live *fcpA⁻* mutant conferred complete protection against death after heterologous infection on both hamster and mouse models. This result is highlighted by the poor performance of the heat-killed vaccine when given in a single dose. Given the promising results obtained with a single dose of the attenuated-vaccine, a boost dose was not evaluated. However, it should be considered for future experiments. Furthermore, we observed that the cross-protection against death after vaccination with the *fcpA⁻* mutant was dose dependent in the mouse model. In another spirochete, a similar approach has been successfully explored when a single dose of a flagella-less mutant in *Borrelia burgdorferi* induced homologous protection in mouse (*Sadziene et al., 1996*). More recently, a target mutant in *B. burgdorferi* induced cross immunity, but four immunizations were necessary for full protection (*Hahn et al., 2016*). Previous work with attenuated-vaccine in Leptospira showed evidence of protection against homologous challenge in guinea-pigs, cattle, hamsters, and swine, but only partial cross-protection in hamsters and gerbils (*Adler, 2015*; *Srikram et al., 2011*). To our knowledge, this is the first study showing evidence of cross-protection across species with a vaccine in leptospirosis.

A single high-dose of the *fcpA⁻* mutant induced sterilizing immunity in the mice model of leptospirosis. Other than reduction of lethality, the ability to prevent transmission is a key parameter to determine the success and feasibility of a leptospiral vaccine. Whole-cell vaccines have shown divergent levels of protection against colonization in different species (*Sonada et al., 2018*; *Murray et al., 2013*; *Clough et al., 2018*; *Bolin and Alt, 2001*). In our hamster model we observed a significant reduction of the bacterial burden in the kidney of vaccinated animals, without significant levels of protection. Mice models are good alternatives for sublethal leptospirosis infection and particularly to study renal colonization, compared to the highly susceptible hamster model (*Gomes-Solecki et al., 2017*). However, the mice model lacks information about susceptibility to a wide range of strains compared to hamsters (*Gomes-Solecki et al., 2017*), and for that reason we were only able to test serovar Manilae as an heterologous model to the serovar Copenhageni attenuated-vaccine. Previous experiments in mice showed that if the challenge dose is not able to cause death of the animals, a portion of the sublethal dose of leptospires is able to escape the blood defenses and colonize the kidneys (*Ratet et al., 2014*). Furthermore, the dose of infection necessary to cause disease in natural infection remains unclear, although recent research showed that the numbers of leptospires available in the environment are low on average (*Casanovas-Massana et al., 2018b*; *Ganoza et al., 2006*). Current animal model experiments focus on unrealistic high challenge doses,

which can devaluate vaccine efficacy estimation. Despite that, our results indicate that the *fcpA*⁻ attenuated-vaccine is able to induce a robust sterilizing immunity capable of preventing death and colonization after a high lethal dose of infection.

Anti-Leptospira protein antibodies, and not agglutinating antibodies, are the correlates of *fcpA*⁻ attenuated-vaccine-mediated cross-protective immunity. The essential role of antibodies for protection against leptospirosis has been long described (*Adler et al., 1980*; *Inada et al., 1916*). However, the key knowledge gap for vaccine development is whether naturally acquired infection elicits immunity to reinfection. There have been no attempts to address this question in either humans or animals and our understanding has thus relied solely on experimental models of acquired immunity. Passive transfer experiments have shown that whole-cell vaccine immunity is antibody-mediated (*Inada et al., 1916*; *Adler, 2015*). Serovar-specific protection correlates with agglutinating antibody titers (*Adler et al., 1980*; *Chapman et al., 1991*), which are directed against lipopolysaccharides (LPS) (*Challa et al., 2011*). Anti-leptospiral LPS antibodies are generated by whole-cell based vaccines, and the restricted protection these vaccines confer is one of their major limitations (*Adler, 2015*). Recently, immunization with an attenuated, LPS-deficient strain of serovar Manilae conferred protection against infection with serovar Pomona and other *L. interrogan*s species, suggesting that anti-Leptospira protein responses are indeed cross-protective (*Srikram et al., 2011*; *Murray et al., 2018*). Our results not only corroborate those previous studies, but also shows strong evidence, through passive transfer experiments, that the cross-protection mediated by anti-protein antibodies is transferable.

Comprehensive high-throughput analysis of the antibody response identified potential protein candidates with a major role on the cross-protection conferred by live-attenuated-vaccine. The attenuated-vaccine induced a robust anti-Leptospira protein antibody response, but these antibodies recognized a limited repertoire of 154 antigens. Although the numbers in our study are higher compared to the 11 proteins reported by a similar study (*Srikram et al., 2011*), this difference likely relates to the significantly higher sensitivity of the proteome array compared to the 2-DIGE method used previously. Nevertheless, our approach was validated by the enrichment analysis of the reactive proteins, which showed that proteins classified as OMP and cytoplasmic membrane-associated, and proteins predicted to be involved in cell wall/membrane and intracellular trafficking/secretion were highly enriched among the selected proteins. Moreover, using two animal models and different doses of the vaccine allowed us to reduce this number to 41 selected proteins, characterizing correlates of immunity. By implementing a systematic approach to identifying cross-protective antigens, we address a critical barrier to exploiting this database of genomic information for vaccine discovery.

Given the limitations of the whole-cell vaccines currently in use, identifying the perfect protein candidate has been a major effort in the field of vaccine development for leptospirosis (*Adler, 2015*). Immunization with Lig proteins or LipL32 (*Conrad et al., 2017*; *Ko et al., 2009*; *Humphryes et al., 2014*) has been shown to confer protection, albeit not cross-protective immunity, against experimental infection. However, efficacy results have been proven difficult to replicate (*Deveson Lucas et al., 2011*; *Ko et al., 2009*). As of yet we do not know whether a single protein can be immunoprotective or if the additive value of immunity against a group of proteins is essential for cross-immunity. Lately, studies have shown that a chimeric approach has the potential for an effective universal vaccine against leptospirosis (*Lin et al., 2016*; *Santos and Nascimento, 2018*). Other than Ligs and LipL32, LipL41 has been tested as a vaccine candidate without success (*Ko et al., 2009*) and LIC20172 and LIC11966 have been identified and tested, respectively, as promising candidates for subunit vaccine (*Lata et al., 2018*; *Oliveira et al., 2018*). More recently, a DNA vaccine delivering two leptospiral proteins (RecA and FliD) has shown promising results including high survival rates, sterilizing immunity and heterologous protection, although tested only among two different *L. interrogans* strains (*Raja et al., 2018*). A major limitation for vaccine development for leptospirosis is the lack of evidence whether immunological correlates from experimental animals are relevant to human disease. However, our results suggest that the same antigens that confer cross-protective immunity in the attenuated-vaccine model may play a role in naturally acquired immunity in humans.

This study is the first description of vaccine that elicits cross-protection against disease caused by different species of the genus Leptospira, and that the protection is transferable through anti-protein antibodies. Effective prevention is urgently needed as drivers of disease transmission continue to intensify, like rapid urbanization and climate change. The *fcpA*⁻ mutant induced robust and cross-

protective immune responses against protein moieties conserved across pathogenic Leptospira. We identified a small repertoire of biologically relevant proteins that can be used in the field of vaccine and diagnostic development for leptospirosis. Characterization of these proteins as novel putative virulence factors can also provide new insights on the pathogenesis of the disease. Both approaches are currently being tested by our group as we try to identify an absolute correlate of immunity for leptospirosis. Nonetheless, these findings indicate that a single dose of *fcpA*⁻ mutant elicits cross-protective immunity against serovars belonging to species of Leptospira that encompasses the majority of serovars of human and animal health importance. Taken together, those findings highlight the feasibility to use this model to formulate novel approaches for prevention, such as a synergistic development of vaccines for human and animal health in a diverse range of epidemiological settings.

## Materials and methods

**Key resources table**

| Reagent type (species) or resource | Designation | Source or reference | Identifiers | Additional information |
|---|---|---|---|---|
| Strain, strain background (*Leptospira interrogans*) | Fiocruz L1-130 *fcpA*⁻ mutant | *Wunder et al., 2016a* doi:10.1111/mmi.13403 | *L. interrogans* serovar Copenhageni strain Fiocruz L1-130 *fcpA*⁻ | Fiocruz L1-130 *fcpA*⁻ mutant |
| Antibody | Peroxidase AffiniPure Goat Anti-Mouse IgG (H+L) | Jackson Immuno Research | Cat# 115-035-166 RRID:AB_2338511 | ELISA (1:50,000) WB (1:100,000) |
| Antibody | Peroxidase AffiniPure Goat Anti-Mouse IgG (H+L) | Jackson Immuno Research | Cat# 107-035-142 RRID:AB_2337454 | WB (1:100,000) |
| Antibody | Hamster α-*fcpA*⁻ sera (Hamster polyclonal) | This paper | | See Material and methods WB (1:100) |
| Antibody | Hamster control sera (Hamster polyclonal) | This paper | | See Material and methods WB (1:100) |
| Chemical compound, drug | SureBlue Reserve TMB 1-Component Microwell Peroxidase Substrate | SeraCare | Cat. #: 5120–0081 | |
| Chemical compound, drug | Sulfuric Acid ($H_2SO_4$) | Sigma-Aldrich | Cat. #: 258105 | |
| Chemical compound, drug | Proteinase K | Thermo Fisher | Cat. #: EO0491 | |
| Chemical compound, drug | Platinum Quantitative PCR Supermix-UDG | Thermo Fisher | Cat. #: 11730017 | |
| Software, algorithm | GraphPad Prism software | GraphPad Software | RRID:SCR_015807 | Version 8.0.0 |
| Software, algorithm | RStudio software | RStudio | RRID:SCR_000432 | Version 1.0.153 |

### Vaccine and challenge strains

Leptospires were cultivated in liquid EMJH medium (*Johnson and Harris, 1967*) supplemented with 1% rabbit serum. *Leptospira interrogans* serovar Copenhageni strain Fiocruz L1-130 *fcpA*⁻ mutant (*Wunder et al., 2016a*) and all the seven different strains used for the challenge experiments (*Supplementary file 1*) were incubated up to 7 days at 29°C, till they reached log phase (between 4 and 5 days of culture). For all immunization or infection experiments, the correct number of Leptospira was determined by a Petroff-Hausser counting chamber (Fisher Scientific).

The heat-killed vaccine was prepared by heat-inactivating preparations of *L. interrogans* strain Fiocruz L1-130 at 56°C for 20 min.

### Animal experimentation
#### Dissemination studies

For the dissemination experiments with the Fiocruz L1-130 *fcpA*⁻ mutant and L1-130 heat-killed vaccine in hamsters, a group of fifteen 3-week-old male Golden Syrian hamsters (Envigo) was inoculated

subcutaneously with a dose of $10^7$ leptospires in 0.5 mL of EMJH medium. A group of three animals was euthanized at 1, 4, 7, 14, and 21 days after infection. As a control, a group of nine animals was infected with Fiocruz L1-130 WT using the same route and dose, and animals were euthanized at days 1 and 4 after infection. The final group was euthanized at onset of disease. After euthanizing the animals, blood, kidney, liver, and brain were carefully removed, collected into cryotubes, and immediately placed into liquid nitrogen before being stored at −80°C until extraction of DNA. Kidney and blood were inoculated in EMJH for culture of leptospires when necessary.

For the experiment in mice, groups of three 4-week-old female C57BL/6 mice (Jackson laboratory) were inoculated subcutaneously with different doses of the vaccine ($10^7$, $10^5$, $10^3$, $10^2$, and $10^1$) and a control group with three animals was inoculated with Fiocruz L1-130 WT with a dose of $10^7$ leptospires. Blood was collected by retro-orbital bleeding at 1, 4, 8, 13, 15, 18, and 21 days after infection.

## Immunization and challenge

All vaccination experiments (*Figure 2A*) were performed using 3-week-old male Golden Syrian hamsters (Envigo) or 4-week-old female C57BL/6 mice (Jackson laboratory), divided into groups of six to nine or four to eight animals, respectively. Animals were vaccinated with Fiocruz L1-130 *fcpA⁻* mutant using the subcutaneous route. Hamsters were vaccinated with a single dose of $10^7$ leptospires and mice were vaccinated with a range of doses ($10^7$, $10^5$, $10^3$, $10^2$, and $10^1$) in 500 and 200 μL of EMJH medium, respectively. The heat-killed vaccine was used in a single dose of $10^7$ leptospires by subcutaneous route as a control group in hamster. In addition, groups of animals were injected with phosphate buffered saline (PBS) and served as unvaccinated controls. Blood samples were collected the day before and 20 days post-immunization by retro-orbital bleeding.

Animals were challenged on day 21 post-immunization. Hamsters were challenged by conjunctival inoculation, which mimics the natural route of infection (*Wunder et al., 2016b*; *Adhikarla et al., 2018*) using a lethal dose ($10^8$ leptospires) of a range of serovars whose virulence has been well characterized in our laboratory (*Supplementary file 1*). Hamsters vaccinated with the heat-killed vaccine were only challenged with Fiocruz L1-130 (homologous) or Manilae L495 (heterologous) strains. Whole-cell inactivated vaccines (bacterins) against leptospirosis are known for their lack of cross-reactive protection (*Adler, 2015*), and for that reason we only used one heterologous serovar for comparison. Mice were challenged intraperitoneally with *L. interrogans* serovar Manilae L495 ($10^8$ leptospires) (*Gomes-Solecki et al., 2017*; *Ratet et al., 2014*). After euthanizing the animals, kidneys were collected and stored as described above.

## Passive transfer experiments

Immune sera against Fiocruz L1-130 *fcpA⁻* mutant was generated by immunizing a group of ten 3-week-old male Golden Syrian hamsters using the same protocol, described above. A group of 10 animals injected with PBS was used to obtain control sera. Animals were euthanized at day 21 post-immunization by inhalation of CO. Blood was obtained by cardiac puncture, followed by separation of sera that was subsequently pooled as immune (hamster α-*fcpA⁻*) and control sera.

Immune or control sera were passively transferred to groups of 5 naïve female mice and seven naïve male hamsters (6–7-week-old) in a dose of 0.5 and 2.0 mL, respectively, using the intraperitoneal route. After 24 hr mice and hamsters were challenged with $10^8$ leptospires *of* serovar Manilae L495 (heterologous strain) by intraperitoneal and conjunctival route, respectively, as described above.

## Ethical statement

All animal protocols were approved by the Institutional Committee for the Use of Experimental Animals, Yale University (protocol # 2017–11424). Hamsters and mice were monitored twice daily for endpoints including signs of disease and death, up to 28 days and 14 days post-infection, respectively. Surviving animals at the end of the experiment or moribund animals presenting with difficulty moving, breathing, or signs of bleeding or seizure were immediately sacrificed by inhalation of $CO_2$. Before each blood collection animals were anesthetized by an open-drop method with a mixture of 20% v/v isoflurane in propylene glycol.

## Serology

Pre- and post-vaccination sera were obtained by centrifugation of clotted blood at 1000 g for 15 min at room temperature. Sera samples were kept frozen at −20°C until analysis for the presence of antibodies against leptospires by MAT, ELISA, immunoblotting, and proteome array.

MAT was performed using standard practices and as previously described (*Ko et al., 1999*). Serum was diluted at 1:100 and tested against all the strains used in this project (*Supplementary file 1*). Positives samples were subsequently titrated.

For the ELISA, whole cell lysate was prepared by centrifugation of *L. interrogans* serovar Manilae L495 and Fiocruz L1-130 cultures ($10^8$ cells) at 12,000 rpm, 4°C for 20 min. The pellets were washed twice with PBS and resuspended in 500 µL of PBS. Resuspended cultures were sonicated in ice for 6 cycles at 30 kHz with a power output of 300 W. Lysates were quantitated by Bradford assay and employed as antigen at a concentration of 150 ng/well (in 0.05 M carbonate buffer, pH 9.6). Flat-bottomed polystyrene microtiter plates (Corning) were coated with Leptospira antigen and incubated overnight at 4°C. The plates were washed three times with PBS-0.05% (vol/vol) Tween 20 (PBST) and incubated with blocking solution (5% blocking milk in 2% [wt/vol] bovine serum albumin) for 2 hr at 37°C. After four washes with PBST, wells were incubated with mouse immune sera, diluted 100-fold in 2% BSA, for 1 hr at 37°C. Secondary anti-mouse HRP conjugated antibody (Jackson ImmunoResearch) was used at a dilution of 50,000 (2% BSA) and incubated for 1 hr at 37°C. TMB SureBlue Reserve (SeraCare) was used for detection and the reaction was stopped by adding 100 µL of 2 N $H_2SO_4$. Absorbance (450 nm) was recorded by microplate reader (Biotek).

To evaluate the effect of proteolytic enzyme treatment on Leptospira antigen we used the protocol previously described (*Udaykumar and Saxena, 1991*). Briefly, Leptospira antigen coated in assay wells was treated with 0.1 mg of Proteinase K (Invitrogen) at 37°C for 2 hr. The plates were washed three times with PBST to remove unbound proteins and followed by blocking and testing as described above.

## qPCR

DNA was extracted from blood and tissue samples using the Maxwell16 (Promega Corporation) instrument following the manufacturer's instructions. Quantitative Real-time PCR assays were performed on hamster and mouse tissues using an ABI 7500 instrument (Applied Biosystems) and Platinum Quantitative PCR Supermix-UDG (Invitrogen Corporation) with *lipL32* primers and probe as described previously (*Wunder et al., 2016b*).

## Western blot

Immunoblots with whole cell extract of Leptospira strains were performed as previously described (*Lourdault et al., 2011*). Western blot was performed with a pool of hamster or mice immune sera α-*fcpA*⁻ at dilution of 1:100. For subsequent detection, HRP goat anti-mouse or anti-hamster's serum (Jackson ImmunoResearch) was employed at dilution of 1:100,000. Blots were analyzed using Chemi-Doc Imager (Bio-Rad).

## Proteome array

The full ORFeome was amplified from *Leptospira interrogans* serovar Copenhageni strain Fiocruz L1-130 as previously described (*Lessa-Aquino et al., 2013*; *Lessa-Aquino et al., 2015*). The ORFs larger than 150 bp were amplified from genomic DNA, followed by recombination cloning into a T7 expression vector. Genes larger than 3 kb were cloned as segments. A list of 660 most reactive antigens were selected from previous studies with human sera of patients with leptospirosis (*Lessa-Aquino et al., 2013*; *Lessa-Aquino et al., 2017*; *Lessa-Aquino et al., 2015*) and used for the hamster experiments. Mouse sera were tested in an array containing 330 proteins selected based on the latter. Proteins were expressed in the in vitro transcription/translation (IVTT) RTS 100 *E. coli* HY system (5 PRIME) and synthesized crude proteins were printed on 3-pad nitrocellulose-coated AVID slides (Grace Bio-Labs) using a Gene Machine OmniGrid 100 microarray printer (Genomic Solutions). In addition to IVTT expressed proteins, each array contained no DNA control spots consisting of IVTT reactions without the addition of a plasmid, serial dilutions of purified IgG/spots.

The arrays were probed for IgG reactivity. For serum samples, the arrays were probed at 1/100 dilution in protein array blocking buffer (GVS) supplemented with *E. coli* lysate (Genscript) at a final

concentration of 10 mg/mL to block anti-*E. coli* antibodies. The arrays were incubated overnight at 4°C with constant agitation. After the overnight incubation, the arrays were washed three times with T-TBS and then incubated for 45 min at RT with biotin-conjugated anti-human IgG secondary antibody (Jackson ImmunoResearch), diluted at 1/400 in array blocking buffer, followed by Qdot 800 streptavidin conjugate (ThermoFisher Scientific). The arrays were air dried after brief centrifugation. IgG signals were detected with ArrayCam 400 s Microarray Imaging System (Grace Bio-Labs) for Q800. The array signal intensities were quantified using QuantArray software. Mean pixel intensities are corrected for local background of all spots. Protein expression was validated by microarray using monoclonal anti-polyhistidine (clone His-1, Sigma).

In addition, 30 sera (acute and convalescent) of human patients from Salvador, Brazil with confirmed acute leptospirosis were probed in the array containing 660 proteins as described above. Patient samples were collected and selected as previously described (*Lessa-Aquino et al., 2013*).

## Data analysis

We analyzed the $\log_{10}$ fold change (LFC) between pre- and post-vaccination proteome signal intensities. We subtracted the chip background and set the negative values to one (to avoid issues taking logarithms) before calculating the LFC. Analyses were conducted on three data sets: the hamster data, which used a single attenuated-vaccine dose of $10^7$ leptospires (Hamster $10^7$), the mouse dose–response data including all vaccine doses $10^1$, $10^2$, $10^3$, $10^5$, and $10^7$ (Mouse all), and the subset of mice given a dose of $10^7$ leptospires of the attenuated-vaccine (Mouse $10^7$). The decision of those analysis group was made based on the fact that the high dose of the vaccine in mouse (Mouse $10^7$) was the same as in the hamster experiments (Hamster $10^7$) and was the only dose in mouse that gave 100% protection against death and colonization. Furthermore, the other doses (Mouse all) provided different levels of protection when combining death and colonization and the comparison analysis of all those three groups would increase our chances to identify potential protein targets with a role in those outcomes.

Exploratory analysis of mouse data showed a dose–response relationship, with increased vaccine dose associated with increased mean signal intensity (*Figure 4—figure supplement 2*) as well as decreased death and colonization. We used a model that allowed us to quantify this dose–response relationship when present and to instead measure the contrast between vaccinated and unvaccinated animals if only a single dose was used. Each antigen was modeled separately. For each antigen, we fit a linear model for the LFC in each animal $A$ as $\mathrm{LFC}_A = \mathrm{Experiment}_A + V_A + \mathrm{LogDose}_A$ where experiment was a factor on four levels for the mice and two levels for the hamsters. $V_A$ is an indicator variable for whether the animal $A$ received the attenuated-vaccine or a control injection. These terms were included in all models. The LogDose term was only included in the analysis of the dose–response relationship in mice and is the logarithm of the dose (0 for control animals, 1, 2, 3, 5, and 7). The indicator variable $V_A$ prevents LogDose = 0 for control animals from being treated as a true zero. Our statistic of interest was the t-statistic. For the Hamster and Mouse $10^7$ models we interpreted the $V_A$ t-statistics, and for the Mouse All dose–response model we interpreted LogDose t-statistics. We used the Benjamini–Hochberg (BHp) correction (*Benjamini and Hochberg, 1995*) to control the false discovery rate at 0.05. This analysis was conducted in RStudio software (RStudio).

Prism 8 (GraphPad Software) was employed for all the statistical analysis of in vivo data. Fisher's exact test and analysis of variance (ANOVA) were applied to assess statistical differences between pairs of groups and multiple groups, respectively. A p-value of <0.05 was considered significant. A binomial proportion confidence interval was calculated to determine the efficiency of the vaccine in both hamster and mouse. Protein homologies of *L. interrogans* serovar Copenhageni strain Fiocruz L1-130 were identified by a BLAST search (http://www.ncbi.nlm.nih.gov/BLAST/). Clusters of ortholog groups (COGs), pSortB localization, transmembrane domains (TMhmm), and signal peptide (SignalP) information were obtained from Genoscope platform (http://www.genoscope.cns.fr/agc/microscope/home/). p-Value for enrichment statistical analysis was calculated using Fisher's exact test in the R environment (http://www.r-project.org).

## Acknowledgements

We would like to thank the Leptospirosis team at Instituto Gonçalo Moniz, Oswaldo Cruz Foundation, Salvador, Brazil, for all the support and technical assistance.

This work was supported by NIH grants R01AI052473, U01AI088752, R01TW009504, and R01AI121207 (AIK). CH was supported by Programa Ciências sem fronteiras, CNPq, Brazil.

## Additional information

### Funding

| Funder | Grant reference number | Author |
|---|---|---|
| National Institutes of Health | R01AI052473 | Elsio A Wunder<br>Albert Ko |
| National Institutes of Health | U01AI088752 | Elsio A Wunder<br>Haritha Adhikarla<br>Albert Ko |
| National Institutes of Health | R01TW009504 | Elsio A Wunder<br>Albert Ko |
| National Institutes of Health | R01AI121207 | Elsio A Wunder<br>Li Liang<br>Philip L Felgner<br>Albert Ko |
| Brazilian Council for Scientific and Technological Development | Science without Borders | Camila Hamond |

The funders had no role in study design, data collection and interpretation, or the decision to submit the work for publication.

### Author contributions

Elsio A Wunder, Conceptualization, Resources, Data curation, Formal analysis, Supervision, Funding acquisition, Validation, Investigation, Visualization, Methodology, Writing - original draft, Project administration, Writing - review and editing; Haritha Adhikarla, Conceptualization, Data curation, Investigation, Methodology, Writing - original draft; Camila Hamond, Investigation, Methodology; Katharine A Owers Bonner, Formal analysis, Methodology, Writing - review and editing; Li Liang, Formal analysis, Investigation, Methodology, Writing - review and editing; Camila B Rodrigues, Vimla Bisht, Investigation; Jarlath E Nally, David P Alt, Resources, Writing - review and editing; Mitermayer G Reis, Resources; Peter J Diggle, Philip L Felgner, Formal analysis, Supervision; Albert Ko, Conceptualization, Resources, Supervision, Funding acquisition, Writing - review and editing

### Author ORCIDs

Elsio A Wunder (ID) https://orcid.org/0000-0002-5239-8511
Katharine A Owers Bonner (ID) https://orcid.org/0000-0002-5323-5079
Li Liang (ID) http://orcid.org/0000-0001-6185-2651
Albert Ko (ID) http://orcid.org/0000-0001-9023-2339

### Ethics

Animal experimentation: All animal protocols were approved by the Institutional Committee for the Use of Experimental Animals, Yale University (protocol # 2017-11424). Hamsters and mice were monitored twice daily for endpoints including signs of disease and death, up to 28-days and 14-days post-infection, respectively. Surviving animals at the end of the experiment or moribund animals presenting with difficulty moving, breathing or signs of bleeding or seizure were immediately sacrificed by inhalation of $CO_2$. Before each blood collection animals were anesthetized by an open-drop method with a mixture of 20% v/v isoflurane in propylene glycol.

### Decision letter and Author response

Decision letter https://doi.org/10.7554/eLife.64166.sa1
Author response https://doi.org/10.7554/eLife.64166.sa2

## Additional files

### Supplementary files

• Supplementary file 1. Leptospira strains used in this study for challenge after vaccination with L1-130 *fcpA*- mutant.

• Supplementary file 2. Efficacy of the immunization with a dose of $10^7$ leptospires of the attenuated L1-130 *fcpA*- mutant in hamsters followed by challenge with $10^8$ leptospires with homologous or heterologous strains by conjunctival route.

• Supplementary file 3. Efficacy of the immunization with different doses of the attenuated L1-130 *fcpA*- mutant in mice followed by challenge with $10^8$ leptospires of heterologous strain by intraperitoneal route.

• Supplementary file 4. Complete list of 154 protein targets identified by the proteome array as correlates of immunity for the attenuated-vaccine model.

• Transparent reporting form

### Data availability

All data generated or analyzed during this study are included in the manuscript and supporting files.

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
