## [Decision Letter]

**Acceptance summary:**

*Leptospira* species are important bacterial pathogens for humans and animals. The authors describe the first vaccine that provides cross protection against different species of *Leptospira*, reduces kidney colonization, but does not provide sterilizing immunity. They also identified potential candidate proteins that could be used for vaccines.

**Decision letter after peer review:**

Thank you for submitting your article "A Live Attenuated Vaccine Model Confers Cross-Protective Immunity Against Different Species of *Leptospira* spp" for consideration by *eLife*. Your article has been reviewed by two peer reviewers, and the evaluation has been overseen by Jos van der Meer as the Senior and Reviewing Editor. The following individual involved in review of your submission has agreed to reveal their identity: Yung-Fu Chang (Reviewer #1).

Below you will find a set of recommendations, which, if completed, would lead to publication in *eLife*.

Summary:

*Leptospira* species are important bacterial pathogens for humans and animals. The authors describe the first vaccine that provides cross protection against different species of *Leptospira*, reduces kidney colonization, but does not provide sterilizing immunity. They also identified potential candidate proteins that could be used for vaccines.Essential Revisions:

The single dose *fcpA*- attenuated vaccine provides great cross protection against death, but only reduces kidney colonization.

1) The authors provided in Supplementary files 2 and 3 a summary of all animal experiments done, but a more detailed table with results of each individual experiment should be added, including control groups that receive PBS.

2) We would like to see the detailed animal experiments in the supplementary data section.

3) Figure 2 should mention that it shows the results of all combined experiments, and the control group should be added to the graphs. The legends of this figure should mention the route of infection for the challenge: SQ for hamsters and IP for mice.

Only one dose of the heat-killed vaccine is used as a control, instead of 2 (subsection “Model for cross-protective immunity to leptospirosis”). Why was not the vaccine administered as commonly used (i.e., 2 doses)? Did the authors try two doses or more of the *fcpA*- vaccine? If so, that could be discussed in the Discussion.

4) In the proteome array, the authors should better explain the rationale for comparing all vaccine doses (mouse) to 10e7 dose (mouse), and not only 10e7 mouse to 10e7 hamster.

5) The Discussion should not start with a new introduction but with the major findings. So start with the last sentence of the first paragraph.

---

## [Author Response]

Essential Revisions:The single dose fcpA- attenuated vaccine provides great cross protection against death, but only reduces kidney colonization.1) The authors provided in Supplementary files 2 and 3 a summary of all animal experiments done, but a more detailed table with results of each individual experiment should be added, including control groups that receive PBS.

We appreciate the suggestion from the reviewer and apologize for not having provided this information. We revised both Supplementary files 2 and 3 and made it more comprehensive, including details about individual experiments and information regarding the PBS control groups.

2) We would like to see the detailed animal experiments in the supplementary data section.

We appreciate the suggestion from the reviewer. As mentioned above, we made a thorough revision of both Supplementary files 2 and 3 to include all the details about individual experiments and information regarding the PBS control groups, but also keeping the overall efficacy of the vaccines.

3) Figure 2 should mention that it shows the results of all combined experiments, and the control group should be added to the graphs. The legends of this figure should mention the route of infection for the challenge: SQ for hamsters and IP for mice.

We thank the reviewer for the comment. We included in the legend of Figure 2 the route of challenged used in those experiments, for both hamsters and mice. We also included a sentence to make it clear that the analysis for efficacy was done using a combination of all experiments.

With regards the request to include control group in the figure, we appreciate the suggestion. However, since the sections A to E of Figure 2 are analysis regarding the efficacy of the vaccine, which was calculated using the results obtained from the PBS control group, that information can’t be included on this figure without compromising its purpose. We hope that the information included in the revised Supplementary files 2 and 3 are sufficient to satisfy the reviewer.

Only one dose of the heat-killed vaccine is used as a control, instead of 2 (subsection “Model for cross-protective immunity to leptospirosis”). Why was not the vaccine administered as commonly used (i.e., 2 doses)?

We appreciate the question from the reviewer. Our goal in this study was to demonstrate the efficacy of the attenuated vaccine after a single dose, given that according to our hypothesis the strain was able to multiply in the host and stimulate the immune system, which would be enough to mount a strong immunity. For that reason and to avoid different doses comparisons we decided to keep a standard protocol of vaccination using only dose for all vaccines. To make this decision clear, we added to the Results section a brief explanation that hopefully will satisfy the reviewer and help the readers: “For the purpose of evaluating the efficacy of the attenuated vaccine after a single dose, we decided to keep a standard protocol for vaccination and thus using only one dose of the heat-killed vaccine as well”.

Did the authors try two doses or more of the fcpA- vaccine? If so, that could be discussed in the Discussion.

Indeed we never tried a boost dose of the attenuated vaccine after the promising results obtained with a single dose. But we acknowledge that for future experiments more doses of the vaccine should be evaluated. We included a sentence in the Discussion section: “Given the promising results obtained with a single dose of the attenuated vaccine, a boost dose was not evaluated. However, it should be considered for future experiments.”

4) In the proteome array, the authors should better explain the rationale for comparing all vaccine doses (mouse) to 10e7 dose (mouse), and not only 10e7 mouse to 10e7 hamster.

We appreciate the reviewers suggestion. We made the decision to consider the analysis of all vaccine doses in mouse given the observation that the protection either for colonization and death in the mouse model was related to the dose of the vaccine. With that analysis, we could identify protein candidates that were potentially involved on those outcomes when performing this combined analysis. We included in the analysis section of the text one sentence to justify the choice: “The decision of those analysis group was made based on the fact that the high dose of the vaccine in mouse (Mouse 10^7^) was the same as in the hamster experiments (Hamster 10^7^) and was the only dose in mouse that gave 100% protection against death and colonization. Furthermore, the other doses (Mouse all) provided different levels of protection when combining death and colonization and the comparison analysis of all those three groups would increase our chances to identify potential protein targets with a role in those outcomes.”

5) The Discussion should not start with a new introduction but with the major findings. So start with the last sentence of the first paragraph.

We appreciate the reviewer suggestion. We agree that the suggestion improves the message, and we shifted the order of the paragraph starting with the sentence mentioned by the reviewer.